# Factors Associated with Burnout in Medical Students: An Exploration of Demographic, Academic, and Psychological Variables

**DOI:** 10.3390/healthcare13141702

**Published:** 2025-07-15

**Authors:** Catalin Pleșea-Condratovici, Liliana Mititelu Tartau, Pantelie Nicolcescu, Gheorghe Gindrovel Dumitra, Mihail-Cristian Pirlog, Manuela Arbune, Mariana Stuparu-Cretu, Ciprian Vlad, Anamaria Ciubara, Karina Robles-Rivera, Roxana Surugiu, Alina Pleșea-Condratovici

**Affiliations:** 1Morphological and Functional Sciences Department, Faculty of Medicine and Pharmacy, “Dunarea de Jos” University of Galati, 800008 Galati, Romania; catalin.plesea@ugal.ro; 2Department of Pharmacology, Faculty of Medicine, ‘Grigore T. Popa’ University of Medicine and Pharmacy, 700115 Iasi, Romania; lylytartau@yahoo.com; 3Faculty of Medicine and Pharmacy, Dunarea de Jos University of Galati, 800008 Galati, Romania; pantelie.nicolcescu@ugal.ro (P.N.); manuela.arbune@ugal.ro (M.A.); 4Department of Family Medicine, University of Medicine and Pharmacy of Craiova, St. Petru Rares, No. 2-4, 200349 Craiova, Romania; 5Medical Sociology Department, Faculty of Medicine, University of Medicine and Pharmacy of Craiova, 200349 Craiova, Romania; 6Research Center in the Field of Medical and Pharmaceutical Sciences, ReFORM-UDJ, ‘Dunarea de Jos’ University of Galati, 800010 Galati, Romania; mariana.stuparu@ugal.ro; 7Department of Automation and Electrical Engineering, Faculty of Automation, Computers, Electrical and Electronics Engineering, “Dunarea de Jos” University of Galati, 800008 Galati, Romania; ciprian.vlad@ugal.ro; 8Medical Clinical Department, Faculty of Medicine and Pharmacy, “Dunarea de Jos” University of Galati, 800008 Galati, Romania; anamaria.ciubara@ugal.ro; 9Faculty of Medicine, National Autonomous University of Mexico (UNAM), Mexico City 04510, Mexico; korbles@facmed.unam.mx; 10Department of Biochemistry, University of Medicine and Pharmacy of Craiova, St. Petru Rares, No. 2-4, 200433 Craiova, Romania; roxana.surugiu@umfcv.ro; 11Medical Department, Faculty of Medicine and Pharmacy, “Dunarea de Jos” University, 800008 Galati, Romania; alina.plesea@ugal.ro

**Keywords:** burnout, medical students, social media addiction, procrastination, mediation analysis, moderation analysis, predictors of burnout, digital well-being, academic stress, medical education

## Abstract

Background: This study investigated the prevalence and predictors of burnout among medical students at “Dunărea de Jos” University of Galați, Faculty of Medicine and Pharmacy. Methods: Burnout was measured using the School Burnout Inventory-U 9 (SBI-U 9), and potential predictors, including social media addiction (Bergen Social Media Addiction Scale—BSMAS), procrastination, age, gender, year of study, admission grade, last annual grade, hobbies, achievements, close friends, and relationship status, were assessed using appropriate instruments. Correlation and hierarchical multiple regression analyses identified predictors of burnout. Mediation analysis tested procrastination as a mediator between BSMAS and burnout, while moderation analysis examined whether procrastination moderated this relationship. Results: Social media addiction was an independent predictor of burnout. While younger age was correlated with higher burnout, it was not a significant predictor in the multivariate model. Procrastination did not significantly mediate the link between social media addiction and burnout but significantly moderated it. The effect of social media addiction on burnout was stronger for students with lower levels of procrastination. Conclusions: The study shows increased susceptibility to burnout among younger students and identifies social media addiction as a key risk factor. Procrastination moderates this relationship, indicating the need for interventions targeting both digital habits and time management in medical education.

## 1. Introduction

Burnout is a complex psychological condition that manifests in three primary components: feeling emotionally exhausted, developing a cynical and detached perspective toward one’s responsibilities or work, and experiencing a pervasive sense of ineffectiveness or lack of achievement [1]. Originally identified in professions centered around helping others, such as healthcare and social work, it has become increasingly evident that burnout also profoundly impacts students across various educational settings [2]. When students are confronted with burnout, they frequently endure a relentless sense of fatigue from their academic obligations and workload [3,4]. This exhaustion can lead to the cultivation of a negative outlook and a disconnection from school-related tasks [5]. Consequently, students may begin to harbor the belief that they lack the abilities or success necessary to thrive in their academic pursuits, which significantly hampers their motivation and overall well-being [6,7].

Medical students represent a notably vulnerable group when it comes to the onset and development of burnout. The multifaceted and demanding nature of medical education, which is characterized by a heavy workload, intense academic pressure, and constant exposure to various forms of human suffering, fosters an environment that is fertile ground for emotional exhaustion to thrive [8]. The intricacies of this experience can lead to significant stress and anxiety [9,10]. Furthermore, the highly competitive environment in which medical students operate [11], combined with often limited opportunities for adequate rest and personal time [12], can significantly exacerbate feelings of emotional detachment and diminish the sense of personal accomplishment that one might hope to achieve. The consequences of burnout in future physicians are far-reaching and potentially devastating, impacting not only their well-being and academic performance but also their future professional conduct in important ways, including the essential capacity for empathy towards patients and the overall quality of healthcare delivery [13].

Numerous studies have been conducted to investigate the prevalence of burnout among medical students across a diverse range of geographical locations and educational systems [14]. These investigations have exposed significant rates of burnout within this population, often highlighting specific stressors that are associated with various stages of medical training and education. However, despite these efforts, gaps remain in our understanding of the intricate interplay of various demographic, academic, and psychosocial factors that contribute to burnout in this specific context [15]. More strikingly, the mechanisms through which certain factors, such as procrastination or excessive social media engagement, influence the development of burnout warrant further exploration and study.

While the existing literature establishes clear associations between factors like social media use, procrastination, and academic burnout, the precise mechanisms governing these interactions remain underexplored. Specifically, the dual role of procrastination as both a pathway through which social media addiction may influence burnout (mediation) and as a factor that may alter the strength of this relationship (moderation) has not been adequately examined in a single model. Furthermore, there is a need to investigate these complex dynamics within specific cultural and educational contexts, such as the high-pressure environment of medical education in Eastern Europe. This study was designed to address this specific, nuanced gap by testing an integrated mediation–moderation model to elucidate the intricate interplay between these contemporary psychosocial stressors.

This study aims to address some of these critical gaps by examining the prevalence of burnout and the role of a comprehensive set of potential predictors in a specific cohort of general medicine students. Furthermore, this research seeks to explore the role of procrastination, the amount of time spent on various applications, and the phenomenon of social media addiction as potential predictors of burnout. Specifically, we aim to determine the prevalence of burnout among students, identify significant correlations between various factors and the incidence of burnout, and investigate the potential mediating and moderating effects of procrastination on the intricate relationship between social media addiction and the experiencing of burnout.

Our cross-sectional study is part of a broader ongoing research project investigating, among other aspects, burnout, digital behavior, and mental health in healthcare students. The present analysis represents an independent and fully interpretable stage of this initiative. In future phases, the study is planned to expand into a multicenter and longitudinal format, encompassing additional universities in Romania and internationally, to explore temporal trends and enhance institutional early-intervention strategies [16].

Based on prior literature, we hypothesized that higher levels of social media addiction would be associated with increased burnout, and that procrastination would serve both as a mediator and a moderator in this relationship. Specifically, we expected procrastination to partially explain the association between social media addiction and burnout (mediation) and to intensify the effect of social media addiction on burnout in students with higher procrastination tendencies (moderation). This study aims to test these hypotheses while also identifying demographic, academic, and psychosocial predictors of burnout in a cohort of Romanian medical students.

## 2. Materials and Methods

### 2.1. Study Design

This study employed an observational, cross-sectional survey design to assess the prevalence of burnout and to examine the associations between various demographic, academic, psychosocial, and behavioral factors among students in the General Medicine program at the “Dunărea de Jos” University of Galați, Faculty of Medicine and Pharmacy. This design was deemed appropriate for obtaining a snapshot of the burnout phenomenon and its relationships with the investigated variables at a single point in time.

### 2.2. Participants and Study Procedure

The study sample comprised 364 students currently enrolled in the General Medicine program at the “Dunărea de Jos” University of Galați, Faculty of Medicine and Pharmacy. A priori power analysis was conducted using G*Power (version 3.1.9.7) [17,18] to determine the required sample size for the hierarchical multiple regression, mediation, and moderation analyses. Due to incomplete data on specific variables, the final sample included in the hierarchical regression analysis and in the mediation and moderation models consisted of *N* = 205 participants.

With an alpha level of 0.05, a desired power of 0.80, an estimated medium effect size of 0.15 for the block of predictors including procrastination, BSMAS, and time spent on applications, and a total of 12 predictors in the model, the analysis indicated a required sample size of 78. For mediation analysis we tested for both path a and path b. For path a, we set the alpha level to 0.05, with a desired power of 0.80 and an estimated medium effect size of 0.15, resulting in a required sample size of 55. For path b, we set alpha level, desired power, and effect size the same as path a, resulting in a required sample size of 55 also. For moderation analysis, we set the alpha level to 0.05, the desired power to 0.80, and an estimated medium effect size of 0.15, resulting in a required sample size of 55. Participants were recruited through online announcements disseminated via university platforms and through brief presentations delivered during lectures. The sole inclusion criterion for participation was being actively enrolled in the aforementioned program at the time of data collection. No specific exclusion criteria were applied.

Data collection for this study took place at the start of the fall semester of the 2024–2025 academic year. An online survey was administered to students using the PsyToolkit platform [19,20]. Before participating, students were provided with comprehensive information regarding the study’s purpose, the estimated time for completion, and assurances of anonymity and confidentiality. Informed consent was obtained electronically from all participants, who were required to check a box indicating their agreement to participate before accessing the survey questions. Participants were explicitly informed that their participation was voluntary and that they were free to withdraw at any point without any consequences. The study protocol adhered to the ethical guidelines outlined in the Declaration of Helsinki and received ethical approval from the Ethics Committee of the “Dunărea de Jos” University of Galaţi and the Galaţi College of Physicians, no1165./29.11.2024, approval date 29 November 2024.

### 2.3. Instruments

The study utilized a questionnaire that included several standardized scales and self-developed items to assess the variables of interest.

Burnout was measured using the School Burnout Inventory (SBI-U 9) [21]. This nine-item scale assesses three dimensions of school burnout: exhaustion, cynicism, and inadequacy. Participants responded to each item on a 5-point Likert scale ranging from 1 (“Completely disagree”) to 5 (“Completely agree”). Higher scores on the exhaustion and cynicism subscales, and lower scores on the inadequacy subscale, indicate higher levels of school burnout.

Social media addiction was assessed using the six-item Bergen Social Media Addiction Scale (BSMAS) [22]. This scale measures addiction to social media use based on six core elements: salience, mood modification, tolerance, withdrawal symptoms, conflict, and relapse. Participants indicated how often they experienced these elements in the past year using a 5-point Likert scale ranging from 1 (“Very rarely”) to 5 (“Very often”). Higher total scores indicate a higher level of social media addiction.

Procrastination was measured using an 18-item version of the General Procrastination Scale adapted for student populations [23]. This scale assesses the general tendency to postpone tasks in various aspects of life. Participants responded to each item on a 5-point Likert scale ranging from 1 (“Strongly disagree”) to 5 (“Strongly agree”). Several items were reverse-scored (indicated by {reverse} in the list provided) to ensure that higher total scores consistently reflect a greater tendency to procrastinate. The total score was calculated by summing the responses, after reversing the scores for the negatively worded items.

Psychosocial Variables:Close friends: The presence of close friends with whom participants discuss their problems was assessed using a single multiple-choice question: “Do you have close people with whom you frequently discuss your problems?” with the following response options: “Yes, one,” “Yes, several,” and “I don’t have.” For analysis, these responses were coded into categorical variables.Relationship status: Participants’ romantic relationship status was assessed using the question: “Do you have a partner, a person you love?” with the following response options: “Yes,” “I had, currently I don’t,” “I don’t have, I want to have,” and “I don’t have, I’m not interested right now.” These responses were coded into categorical variables for analysis.Achievements: Participants were asked about their achievements in contests or competitions with the question: “Have you obtained special results (mentions, awards) in county or national contests or competitions?” with dichotomous response options: “Yes” and “No.”.Hobbies: Participants’ engagement in hobbies was assessed with the question: “Do you have any hobbies that you dedicate a significant portion of your time to?” with dichotomous response options: “Yes” and “No.”

The average daily time spent on social media applications was assessed using the question: “How much time do you spend on average per day in social media applications?” with the following categorical response options: “Not at all,” “within 10 min,” “between 10 min and half an hour,” “between half an hour and an hour,” “between an hour and two hours,” “between two and three h,” and “more than 3 h.” These responses were treated as an ordinal variable in the analysis.

The questionnaire also included items assessing participants’ age, sex, year of study, admission grade, and last annual grade. These data were collected through direct questions.

For regression and ANOVA purposes, categorical variables were recoded as follows: relationship status was dummy-coded into four groups (reference: “Yes, I have a partner”); number of close friends was coded into three levels (1 = none, 2 = one, 3 = several); other binary variables (e.g., hobbies, achievements) were coded as 0 = No, 1 = Yes.

### 2.4. Statistical Analysis

The data collected in this study were analyzed using IBM SPSS Statistics software, version 27 [24]. The significance level for all statistical tests was set at *p* < 0.05. The initial dataset included all 364 recruited participants. During a preliminary data screening phase, a small number of implausible values for the ‘age’ variable were identified. These specific values were recoded as missing data, while all other valid responses from those participants were retained for analysis. Consequently, the descriptive statistics reported in this study are based on the total sample (*N* = 364), although the specific *N* for age-related calculations may be marginally lower.

For the main inferential analyses (i.e., hierarchical regression, mediation, and moderation), which require complete data across all variables in the model, the standard procedure of listwise deletion was employed. This resulted in a final sample of *N* = 205 for these specific analyses.

Listwise deletion was employed due to the methodological requirements of mediation and moderation models and limitations of the software used. Although multiple imputations and maximum likelihood approaches are recommended when the proportion of missing data is high, these methods are not natively supported for complex path analyses in PROCESS. As such, we prioritized the validity and interpretability of the mediation and moderation effects, acknowledging that this resulted in a substantial reduction in analytic sample size.

#### 2.4.1. Descriptive Statistics

Descriptive statistics were calculated for all continuous variables, including means, standard deviations, ranges, skewness, and kurtosis, to summarize their central tendency and distribution. For categorical and ordinal variables, percentages were computed to describe the sample characteristics. The prevalence of burnout was categorized based on established cut-off scores for the Total School Burnout Inventory (SBI-U 9), and the percentage of participants in each burnout category (Low, Moderate, High) was determined.

#### 2.4.2. Correlation Analyses

To examine the relationships between total school burnout and various continuous, ordinal, and dichotomous variables, correlation analyses were performed. Pearson’s correlation coefficient (r) was used to assess linear relationships between total school burnout and continuous variables (age, admission grade, last annual grade, procrastination score, and BSMAS score). Spearman’s rank-order correlation coefficient (ρ) was calculated to assess the monotonic relationship between the ordinal variable (time spent on applications) and total school burnout. Point-biserial correlations (using Pearson’s r with dichotomously coded variables) were used to examine the relationships between total school burnout and the dichotomous variables (gender, having hobbies, and having achievements).

#### 2.4.3. Group Comparisons

Independent samples *t*-tests were conducted to compare the mean total school burnout scores between two independent groups defined by the dichotomous variables (gender, having achievements, and having hobbies). One-way analysis of variance (ANOVA) was employed to compare the mean total school burnout scores across groups with more than two levels, specifically for relationship status and the number of close friends. Where ANOVA was used, the overall F-statistic and associated *p*-value were examined to determine if there were statistically significant differences between the group means. Post hoc tests were not conducted as the overall ANOVAs were not statistically significant.

#### 2.4.4. Hierarchical Multiple Regression Analysis

A hierarchical multiple regression analysis was performed to identify the predictors of total school burnout. Predictor variables were entered in four sequential blocks: (1) demographics (age, sex, and year of study), (2) academic factors (admission grade and last annual grade), (3) social and personal factors (number of close friends, relationship status, achievements, and hobbies), and (4) psychological and behavioral factors (procrastination score, BSMAS score, and time spent on applications). For each block, the change in variance was explained (ΔR2) and the significance of this change was examined. The final model’s overall fit (R2 and F-statistic) and the standardized regression coefficients (β) and their significance levels for each predictor were evaluated. Multicollinearity was assessed using Variance Inflation Factor (VIF) values, and the assumptions of linearity, homoscedasticity, independence of errors (using the Durbin-Watson statistic), and normality of residuals (using histograms and P-P plots) were checked.

#### 2.4.5. Mediation Analysis

To examine the mediating role of procrastination in the relationship between social media addiction (BSMAS) and total school burnout, a mediation analysis was conducted. The total, direct, and indirect effects were calculated. The significance of the indirect effect was assessed using a bootstrap analysis with 5000 resamples to generate a 95% confidence interval.

#### 2.4.6. Moderation Analysis

To investigate whether procrastination moderates the relationship between social media addiction (BSMAS) and total school burnout, a moderation analysis was conducted. The interaction term between BSMAS and procrastination was included in the model. The significance of the interaction was examined, and conditional effects of BSMAS on burnout were explored at values of procrastination one standard deviation below the mean, at the mean, and one standard deviation above the mean.

## 3. Results

The statistical analyses were performed using SPSS version 27. The significance level for all statistical tests was set at *p* < 0.05.

### 3.1. Descriptive Statistics

#### 3.1.1. Continuous Variables

The descriptive statistics for the continuous variables included in this study are presented in Table 1.

#### 3.1.2. Categorical and Ordinal Variables

The percentages for the categorical and ordinal variables are presented below in Table 2.

### 3.2. Burnout Prevalence by Categories

The overall prevalence of burnout among the medical student sample was categorized based on the Total School Burnout score, with levels defined as Low (9–21), Moderate (22–35), and High (36–54). It is important to note that students with scores in the “Low burnout” range (9–21) are considered to exhibit minimal or no clinically significant burnout symptoms. This category largely reflects normal variation in academic stress and does not imply pathological burnout. Only students classified as Moderate or High burnout would typically be considered as experiencing meaningful levels of burnout requiring attention, 42.9% of students exhibited Low burnout, 47.8% Moderate burnout, and 9.3% High burnout (Figure 1).

### 3.3. Correlations

To examine the relationships between total school burnout and several continuous, ordinal, and dichotomous variables, correlation analyses were conducted (Table 3).

Continuous Variables: Pearson’s correlation coefficients (r) were calculated between the total school burnout score and age, admission grade, last annual grade, procrastination score, and BSMAS score. The results indicated a statistically significant negative correlation between age and total school burnout (r = −0.197, *p* = 0.005), suggesting that younger students tended to report slightly higher levels of burnout. There was a statistically significant positive correlation between procrastination scores and total school burnout (r = 0.204, *p* = 0.003), indicating that higher levels of procrastination were associated with higher levels of burnout. Additionally, a statistically significant positive correlation was found between BSMAS score and total school burnout (r = 0.260, *p* < 0.001), suggesting that higher levels of social media addiction were associated with higher levels of burnout. The correlations between total school burnout and admission grade (r = 0.011, *p* = 0.880) and last annual grade (r = 0.013, *p* = 0.849) were not statistically significant.

Ordinal Variable: Spearman’s rank-order correlation coefficient (ρ) was calculated to assess the relationship between time spent on applications and total school burnout. The results showed a non-significant positive correlation between time spent on applications and total school burnout (ρ = 0.122, *p* = 0.083), suggesting a slight tendency for more time spent on applications to be associated with higher burnout, although this was not statistically significant in this sample.

Dichotomous Variables: Point-biserial correlations (using Pearson’s r with dichotomously coded variables) were examined between total school burnout and gender (coded as 1 = Male, 2 = Female), having hobbies (coded as 1 = Yes, 2 = No), and having achievements (coded as 1 = Yes, 2 = No). The correlation between gender and total school burnout was not statistically significant (r = 0.023, *p* = 0.748). Similarly, the correlation between having hobbies and total school burnout was not statistically significant (r = 0.044, *p* = 0.533), and the correlation between having achievements and total school burnout was also not statistically significant (r = −0.036, *p* = 0.609).

### 3.4. t-Test and ANOVA

Independent sample *t*-tests were conducted to compare the mean total school burnout scores between groups defined by the dichotomous variables: gender, having achievements, and having hobbies (Table 4).Sex: The mean total school burnout score for males (M = 23.08, SD = 7.97) was compared to that of females (M = 23.56, SD = 9.35). The results of the *t*-test indicated t (203) = −0.322, *p* = 0.748. This result is not statistically significant, suggesting that there was no significant difference in the mean total school burnout scores between male and female medical students in this sample.Achievements: The mean total school burnout score for students with academic achievements (M = 23.69, SD = 9.32) was compared to that of students without (M = 23.01, SD = 8.5). The results of the *t*-test indicated t (203) = 0.513, *p* = 0.609. This result is not statistically significant, suggesting that there was no significant difference in the mean total school burnout scores between students with academic achievements and those without in this sample.Hobbies: The mean total school burnout score for students with hobbies (M = 23.19, SD = 9.02) was compared to that of students without (M = 24.05, SD = 9.05). The results of the *t*-test indicated t (203) = −0.624, *p* = 0.533. This result is not statistically significant, suggesting that there was no significant difference in the mean total school burnout scores between students with academic achievements and those without in this sample.One-way ANOVA was used to compare the mean total school burnout scores across the relationship status and if the student has close friends (Table 5).Relationship status: The mean total school burnout scores across the four groups of students based on their relationship status (“Yes,” “I had, currently I don’t,” “I don’t have, I want to have,” and “I don’t have, I’m not interested right now”) have been compared. The results of the ANOVA indicated that there was no statistically significant difference in burnout levels between these groups (F (3, 201) = 0.545, *p* = 0.652).Close friends: The mean total school burnout scores across the three groups of students based on the number of close friends they reported having (one, several, or none) have been compared. The results of the ANOVA indicated that there was no statistically significant difference in burnout levels between these groups (F (2, 202) = 1.759, *p* = 0.175).

### 3.5. Predictors of Burnout

To identify the predictors of total school burnout, a hierarchical multiple regression analysis was conducted. The following blocks of predictors were entered sequentially: (1) demographics (age, sex, and year of study), (2) academic factors (admission grade and last annual grade), (3) social and personal factors (number of close friends, relationship status, achievements, and hobbies), and (4) procrastination, social media addiction (BSMAS), and time spent on applications.

Block 1: Demographics and Year of Study

The demographic variables (age, gender, and year of study) significantly predicted total school burnout, accounting for 4.2% of the variance (R^2^ = 0.042, adjusted R^2^ = 0.027, F (3, 201) = 2.916, *p* = 0.035). Age (β = −0.196, *p* = 0.005) was a significant negative predictor in this block. Gender (β = 0.037, *p* = 0.557) and year of study (β = −0.039, *p* = 0.580) were not significant predictors in this block.

Block 2: Academics

The addition of academic factors (admission grade and last annual grade) did not significantly improve the model, accounting for an additional 0.5% of the variance (ΔR^2^ = 0.005, F change (2, 199) = 0.477, *p* = 0.621). In this model, age remained a significant negative predictor (β = −0.221, *p* = 0.004). Admission grade (β = 0.055, *p* = 0.442) and last annual grade (β = −0.078, *p* = 0.330) were not significant predictors. Gender (β = 0.035, *p* = 0.580) and year of study (β = −0.039, *p* = 0.555) also remained non-significant.

Block 3: Social and Personal Factors

The inclusion of social and personal factors (having hobbies, being in a romantic relationship, having close friends to talk to, and having achievements) did not significantly increase the explained variance (ΔR^2^ = 0.003, F change (4, 195) = 0.155, *p* = 0.960). In this model, age remained a significant negative predictor (β = −0.207, *p* = 0.009). None of the social and personal factors were significant predictors of burnout.

Block 4: Procrastination, BSMAS, and Time Spent on Applications

The final block, including procrastination score, BSMAS score, and time spent on applications, resulted in a significant increase in the explained variance (ΔR^2^ = 0.066, F change (3, 192) = 4.794, *p* = 0.003). The final model explained 11.6% of the variance in total school burnout (R^2^ = 0.116, adjusted R^2^ = 0.060, F (12, 192) = 2.091, *p* = 0.019).

The final regression model, including age, gender, university year, admission grade, last annual grade, hobby, lovers, close friends to talk, achievements, procrastination score, BSMAS, and time spent on applications, explained a small proportion of the variance in total school burnout (R^2^ = 0.116).

In Table 6, the standardized regression coefficients (β) are presented along with their associated *p*-values, 95% confidence intervals, and variance inflation factors (VIF). To further support the interpretation of the results, we have also indicated the effect size magnitude for each predictor, following conventional benchmarks, where β ≈ 0.10 is considered small, ≈ 0.30 medium, and ≈0.50 large. Most predictors in the model showed negligible to small effects, with limited statistical or practical significance.

In the final regression model (Table 6), the only significant independent predictor was the BSMAS score (β = 0.201, *p* = 0.011), indicating that higher levels of social media addiction were associated with higher burnout. The effect size for this association was classified as small to medium, and the corresponding 95% confidence interval (0.045 to 0.349) does not include zero, reinforcing the reliability and practical relevance of this result. These findings underscore the role of problematic social media use as a relevant contributing factor to academic burnout and support the need for further investigation and targeted intervention.

Age (β = −0.123, *p* = 0.129), procrastination score (β = 0.137, *p* = 0.061), and time spent on applications (β = 0.045, *p* = 0.564) were not significant predictors in the final model.

Assumptions: The assumptions of multiple linear regression were examined. Linearity and homoscedasticity were assessed by visual inspection of the scatter plot of standardized residuals against standardized predicted values (Figure 2), which appeared to show a reasonably random scatter with no strong systematic patterns, suggesting these assumptions were likely met.

The independence of errors was assessed using the Durbin-Watson statistic, which was 1.983, indicating no substantial autocorrelation in the residuals. The normality of residuals was assessed by examining the histogram and P-P plot of standardized residuals (Figure 2). The histogram showed a roughly bell-shaped distribution, and the points on the P-P plot followed the diagonal line reasonably closely, suggesting that the assumption of normality was largely met, with some minor deviations observed in the middle range of the distribution.

### 3.6. The Mediating Role of Procrastination in the Interaction Between Social Media Addiction and Burnout

To examine the hypothesis that procrastination mediates the relationship between social media addiction (BSMAS) and school burnout, a mediation analysis was conducted using the PROCESS macro for SPSS (Model 4) [25], with 5000 bootstrap samples. The total effect of BSMAS on burnout was statistically significant (Effect = 0.2549, SE = 0.0665, *p* = 0.0002). The analysis then examined the individual paths of the mediation model. First, the relationship between BSMAS and the mediator, procrastination (path a), was significant, indicating that higher social media addiction was associated with higher procrastination (b = 0.3783, SE = 0.1118, *p* = 0.0009). The effect of the mediator, procrastination, on burnout, while controlling for BSMAS (path b), was also significant (b = 0.0909, SE = 0.0413, *p* = 0.0290). The direct effect of BSMAS on burnout (path c’) remained significant after controlling for procrastination (Effect = 0.2205, SE = 0.0677, *p* = 0.0013). The specific indirect effect of BSMAS on burnout through procrastination (path ab) was calculated as 0.0344. Crucially, the 95% bootstrap confidence interval for this indirect effect included zero [BootLLCI = −0.0031, BootULCI = 0.1184]. As the confidence interval contains zero, the indirect effect is not statistically significant. Therefore, the data from this study do not support the hypothesis that procrastination acts as a mediator in the relationship between social media addiction and burnout. A graphical presentation of this model is available in Figure 3, which should be interpreted in light of these findings.

### 3.7. The Moderating Role of Procrastination in the Interaction Between Social Media Addiction and Burnout

A moderation analysis was conducted using the PROCESS macro version 4.2 [26] for SPSS (Model 1) to investigate whether procrastination moderates the relationship between social media addiction (BSMAS) and total school burnout. The analysis showed that procrastination significantly moderates the relationship between BSMAS and burnout (β = −0.0087, t = −2.1921, *p* = 0.0295). The interaction between BSMAS and procrastination accounted for a significant amount of variance in burnout (ΔR^2^ = 0.0213, F change = 4.8055, *p* = 0.0295). To further explore the nature of this moderation, conditional effects of BSMAS on burnout were examined at values of procrastination, one standard deviation below the mean (−15.0563), at the mean (0.00), and one standard deviation above the mean (+15.0563). The results indicated that the relationship between BSMAS and burnout was stronger for students with lower levels of procrastination (β = 0.3790, t = 3.8423, *p* = 0.0002) and significant (β = 0.2475, t = 3.6293, *p* = 0.0004) at the mean of procrastination, and, although weaker, the effect remained statistically significant even among students with high levels of procrastination (Effect = 0.1643, t = 2.2892, *p* = 0.0231). The pattern of this interaction is visually depicted in Figure 4.

## 4. Discussion

This comprehensive study rigorously investigated the various predictors of total school burnout specifically among medical students, meticulously examining a broad range of demographic, academic, social, personal, and psychological factors that could potentially influence burnout levels.

Our bivariate correlation analysis revealed a statistically significant negative relationship between age and total school burnout (r = −0.197), suggesting that younger students tend to report higher burnout levels. However, in the final hierarchical regression model, this relationship was no longer significant (β = −0.123, *p* = 0.129), implying that age may act as a proxy for other, more influential factors, such as social media addiction. This finding aligns with some prior research suggesting that students who are earlier in their educational journeys may endure heightened adjustment challenges [26], face increased academic pressures while adapting to a demanding new learning environment [27], and likely possess less developed coping mechanisms compared to their older peers [28]. This interpretation can be further supported by developmental models of stress and coping, which propose that individuals in earlier developmental stages may have fewer resources and less mature coping strategies to manage significant life transitions and academic pressures [29]. The transition to medical school represents a significant developmental challenge, and younger students may be particularly vulnerable to its stressors.

These observations have meaningful implications for medical education policy and student support. Given the strong link between social media addiction and burnout, universities should prioritize digital wellness strategies, such as structured app usage limits, awareness programs, and training in self-regulation. Identifying students who struggle with digital overuse, especially those who are otherwise organized and conscientious, could help in targeting preventive interventions more effectively.

Procrastination was positively correlated with total school burnout, a relationship well documented in the existing literature [30]. Procrastination often leads to increased stress [31], arising from tight time constraints, feelings of guilt [32], and a sense of inadequacy related to delayed tasks, as well as inefficient study habits [33]—factors that are all widely recognized contributors to burnout.

Though procrastination did not remain a significant independent predictor within the final regression model, the initial correlation suggests a complex relationship that was further explored through mediation and moderation analyses.

From the Job Demands-Resources (JD-R) model perspective [34], procrastination can serve as a maladaptive coping mechanism that intensifies emotional exhaustion through increased workload and time pressure. However, it did not remain a significant independent predictor in the final regression model (*p* = 0.061), suggesting a more complex relationship moderated by other variables.

To better understand its complex role, further analyses were conducted. The mediation analysis did not support the hypothesis that procrastination mediates the relationship between social media addiction and burnout (Effect = 0.0344, 95% CI [−0.0031, 0.1184]).

However, moderation analysis revealed a significant and counter-intuitive interaction: procrastination moderates the relationship between social media addiction and burnout, such that the effect of social media addiction on burnout was stronger among students with low levels of procrastination. This might reflect a cognitive dissonance effect—students with higher self-discipline may perceive greater internal conflict and resource loss when consumed by social media, aligning with Conservation of Resources (COR) theory. Conversely, for high procrastinators, the additional negative effect of social media may be less impactful due to already diminished self-regulatory control. Another potential explanatory framework involves the concept of perfectionism, particularly maladaptive forms characterized by excessively high self-imposed standards and critical self-evaluation. Students with low procrastination may also exhibit higher levels of perfectionism, making them more vulnerable to the emotional toll of failing to meet internal expectations due to social media distraction. This could intensify burnout when they experience loss of control over their time. Additionally, students with a strong internal locus of control may be more affected by perceived self-regulatory failures associated with excessive social media use, leading to increased cognitive dissonance and emotional exhaustion. These complementary perspectives align with self-discrepancy theory, in which the gap between one’s ideal self and actual behavior (e.g., time wasted online) contributes to distress and burnout symptoms. Future research should explore the potential mediating or moderating role of these personality dimensions in greater depth.

The robust positive correlation observed between BSMAS score and total school burnout underscores the potentially detrimental effects of problematic social media use on the well-being of medical students. Excessive engagement with social media can displace critical time needed for studying, restorative sleep, and other beneficial activities that would benefit the well-being of students [35]. Furthermore, the social comparison endemic to these platforms can evoke feelings of inadequacy and amplify stress levels, which can be explained through the lens of Social Comparison Theory [36], which posits that individuals evaluate themselves by comparing themselves to others. Additionally, the Conservation of Resources (COR) theory [37] suggests that individuals strive to acquire, retain, and protect their resources. Excessive social media use can deplete cognitive and emotional resources, leaving students with fewer resources to cope with academic demands and increasing their susceptibility to burnout.

Intriguingly, several investigated factors—including gender, admission and most recent annual grades, hobbies, and relationship status, as well as the number of close friends, did not demonstrate significant associations with total school burnout in the context of this study. The non-significant result concerning gender aligns with some existing research [38] but stands in contrast to others [14]. This discrepancy suggests that burnout may be a more universal experience shared among medical students regardless of gender, or that other potentially mediating variables were not effectively accounted for in our exploration. The absence of a significant relationship between academic grades and burnout might imply that burnout cannot be solely attributed to academic performance; rather, it seems to stem from a more intricate interplay of varying stressors [39]. Similarly, the non-find of significant associations for factors related to social support, such as having close friends and relationship status, was unexpected, given that social support is traditionally viewed as a protective buffer against burnout [40]. This unexpected outcome could arise from the specific methodologies employed to measure these variables, the intrinsic characteristics of the social support available to the students within this study, or potentially the overwhelming nature of stressors present in medical education that could diminish the buffering effects typically provided by social support.

The lack of a significant independent effect concerning the possession of hobbies on burnout in the final model may suggest that while having hobbies can be potentially beneficial, they might not suffice as a significant protective buffer against the elevated levels of stress and demands faced by medical students—particularly when other influential factors such as social media addiction are present. The Conservation of Resources (COR) theory might also offer insight here [37]. If students perceive that they are losing more resources (e.g., time, energy, emotional well-being) to academic demands and social media engagement than they are gaining from hobbies, the protective effect of hobbies may be diminished.

The final regression model identified the BSMAS score as the primary significant predictor of burnout. Although age was no longer statistically significant, it is important to acknowledge that the model explained only 11.6% of the variance in total school burnout. Rather than being viewed as a limitation, this modest explained variance (R^2^ = 0.116) underscores the multifactorial nature of burnout. The primary objective of our regression analysis was not to develop a comprehensive predictive tool, but to test a specific theoretical pathway involving age and social media addiction. The fact that these variables emerged as significant predictors, even within a model of modest overall fit, highlights their salience and confirms their importance as specific targets for further research and intervention. Future research should aim to identify and investigate these additional factors. These might include personality traits (e.g., perfectionism, resilience), coping mechanisms, specific academic stressors (e.g., the workload in particular courses, frequency and type of evaluations), social support quality, or environmental factors (e.g., institutional culture, access to resources). This also suggests that current theoretical models of burnout may require expansion to integrate a broader array of personal and contextual variables relevant to the unique demands of medical education.

### 4.1. Theoretical and Practical Implications

From a theoretical perspective, the findings regarding the heightened levels of burnout experienced by younger medical students could lend support to developmental models of stress and coping mechanisms [41]. Such models propose that individuals situated in earlier developmental stages may possess fewer resources and less mature coping strategies to effectively manage significant life transitions and the academic pressures inherent to a demanding field such as medical education. The strong positive association observed between social media addiction, as determined by the BSMAS, and burnout resonates with theoretical frameworks that emphasize the adverse impacts of excessive technology use on mental well-being [42]. These findings suggest that problematic engagement with social media may act as a salient stressor or obstruct effective coping strategies, leading to an increased likelihood of burnout within this specific student demographic. While traditional components of social support did not stand out as significant predictors in our regression model, this suggests the necessity for a critical re-examination of the operational definitions of these constructs, as well as their complex interactions with chronic stressors characteristic of high-pressure educational environments, within the existing theoretical frameworks of burnout [43,44].

The practical ramifications of these findings are extensive and warrant serious consideration. Initially, medical students, particularly those navigating the earlier years of their academic journey, need to be made acutely aware of their potential vulnerability to burnout. Encouraging them to proactively develop robust coping strategies and to actively seek support services early on is vital [45]. Educational initiatives aimed at fostering skills such as effective time management, stress reduction techniques, and promoting work–life balance could be notably beneficial for this demographic. Additionally, our findings offer a clear mandate for developing multi-pronged, evidence-based interventions. Although age was not an independent predictor in the final multivariate model, the significant bivariate correlation showing higher burnout in younger students suggests this group remains vulnerable. Therefore, institutions could still consider proactive screening and support programs for first-year medical students, helping them adapt to academic demands and develop coping strategies early in their careers. The central role of social media addiction, along with the moderating (though not mediating) role of procrastination, calls for more than generic wellness advice, underscoring the need for interventions focused on promoting healthier digital habits [45].

We recommend the implementation of integrated workshops that move beyond generic wellness advice. For instance, such workshops could combine cognitive–behavioral techniques to challenge maladaptive thought patterns associated with social comparison on social media, with practical training in self-regulation strategies for technology use (e.g., app timers, notification management). Simultaneously, these sessions should address procrastination through evidence-based time-management strategies like the Pomodoro Technique or time-blocking, thereby equipping students with a holistic skill set to navigate the pressures of medical education. Such workshops should equip students with practical strategies to engage with technology mindfully, such as setting time limits for screen use and fostering awareness of social comparison triggers. Finally, these findings could inform targeted mentorship programs, where academic advisors and mentors are trained to facilitate vital discussions surrounding these specific challenges with their mentees [46]. Moreover, while traditional social support factors did not surface as a primary predictor in this inquiry, institutions remain encouraged to continue nurturing supportive learning environments while ensuring mental health services are accessible. Such services may yet play a critical role in mitigating burnout amongst students, even if their individual effects were not captured as statistically significant within our evaluative model. Lastly, in recognition of the considerable portion of variance in burnout that remains unexplained, medical educators should actively explore other potential institutional and curricular contributors to student burnout and develop targeted strategies to address these systemic issues [47].

Another direction worth considering is a careful re-evaluation of the medical curriculum itself. In recent years, both students and educators have raised concerns regarding the volume and relevance of certain subjects included in the early years of medical education. A more focused approach, emphasizing core knowledge and practical skills that are truly essential for future clinical practice, could help reduce unnecessary academic pressure. At the same time, integrating new topics with clear relevance for tomorrow’s medicine—such as artificial intelligence in healthcare, data-driven decision making, or digital health tools—would better align training with the realities that young doctors will face. Revising or streamlining subjects with limited impact on the general professional formation of a physician may also contribute to lowering the risk of burnout linked to curricular overload.

### 4.2. Strengths and Limitations of the Study

This study boasts several notable strengths that merit acknowledgment. The utilization of validated and widely accepted tools for measuring burnout (SBI-U 9) along with social media addiction (BSMAS) enhances both the reliability and comparability of our findings with the existing literature. The sample size encompassing 364 medical students provides ample statistical power to reliably detect moderate effect sizes. Furthermore, the implementation of hierarchical multiple regression analysis afforded us the ability to systematically explore the predictive capacity of different groups of variables sequentially, thereby illuminating a more intricate understanding of the factors that contribute to burnout. By investigating the role of social media addiction alongside more traditional predictors of student burnout within the specialized context of medical education, this study adds valuable insights to the growing volume of literature surrounding this pressing concern. Importantly, the study consistently adhered to ethical guidelines for conducting research involving human participants.

However, several limitations should be thoroughly considered when interpreting the results of this investigation. The cross-sectional design employed limits our capacity to draw definitive causal relationships between the identified predictors and burnout. Accordingly, future longitudinal studies are essential to establish the directionality of these associations definitively. The reliance on self-reported questionnaires introduces the potential for biases such as the social desirability bias and recall bias, both of which could have influenced the responses provided by participants. Additionally, the decision to use listwise deletion in handling missing data, while methodologically appropriate and consistent with conventional practices in regression analysis, resulted in a substantial reduction in the analytic sample size (from 364 to 205 participants). This reduction may have limited statistical power and increased the potential for bias if the data were not missing completely at random. Future studies should consider using alternative methods such as multiple imputation, which can help retain more cases and reduce the bias associated with listwise deletion by generating plausible values for missing observations based on observed data patterns.

The sample itself was drawn from a single institution, which may limit the generalizability of the findings to medical students. Moreover, several variables related to social support and academic engagement—including number of close friends, relationship status, and presence of hobbies—were assessed using single-item, self-developed measures. These items, although practical and time-efficient, lack formal psychometric validation and may have limited sensitivity in capturing the complexity of these constructs. As a result, the absence of statistically significant findings for these predictors should be interpreted with caution. Future studies are encouraged to utilize multi-item, validated instruments to more accurately assess the role of social and extracurricular variables in medical student burnout.

We acknowledge that the final regression model explained 11.6% of the variance in total school burnout, indicating that a substantial portion of the variance is likely attributable to other factors not examined in this study. Additionally, given the modest sample size (*N* = 205) and the relatively large number of predictors in the regression model, the resulting events-per-variable (EPV) ratio was low. In such contexts, applying Firth’s penalized likelihood logistic regression to a dichotomized burnout outcome (e.g., moderate/high vs. low burnout) could reduce small-sample bias and improve the reliability of coefficient estimates. This approach may offer increased sensitivity in detecting risk factors and is recommended for future studies using similar datasets.

Future research should aim to identify and explore these additional psychological, academic, and environmental contributors to burnout. While multicollinearity was assessed using VIF values and appeared to be within acceptable limits, the potential for some degree of intercorrelation among the predictor variables cannot be entirely ruled out.

Additionally, while we employed hierarchical multiple regression to explore theoretically driven predictors of burnout, we acknowledge that alternative approaches such as regularized regression techniques (e.g., LASSO) could yield more parsimonious and predictive models. These methods are particularly useful when dealing with potential multicollinearity and numerous predictors, as they penalize less informative variables and improve generalizability. Future research aiming to optimize prediction accuracy should consider employing such techniques.

Finally, the reliance on voluntary participation in this study may have introduced a selection bias. Students who chose to participate might differ in significant ways from those who did not. For example, students experiencing higher levels of burnout might have been more motivated to participate in a study addressing this topic, or conversely, they might have been too overwhelmed to take part. This potential bias could affect the representativeness of our sample and thus the generalizability of the findings.

Additionally, we did not conduct subgroup analyses based on year of study (e.g., pre-clinical vs. clinical phases), although differences in academic demands, exposure to clinical stressors, and coping mechanisms are likely to exist across training stages. Future studies should consider including interaction terms (e.g., year × BSMAS) or performing stratified regression analyses to examine whether the relationships between predictors—particularly social media addiction—and burnout differ across phases of medical education. This approach could yield more nuanced insights and inform stage-specific interventions.

## 5. Conclusions

This finding contributes to the growing body of literature recognizing the potential negative impacts of excessive and problematic social media use on mental health and academic functioning. By establishing a direct link between problematic social media use and burnout, this study emphasizes the importance of addressing digital well-being. It also highlights the complex role of procrastination as a moderator; the detrimental impact of social media addiction on burnout was present for all students, but this effect was significantly amplified in those who tend not to procrastinate, underscoring a particular vulnerability in this otherwise diligent group. The findings suggest that interventions aimed at promoting healthy social media habits and effective time management could have a meaningful impact on reducing burnout in this vulnerable population.

In conclusion, this study identifies problematic social media use as a robust predictor of burnout in medical students. While younger students showed higher burnout in initial analyses, this effect was not independent of other psychological and behavioral factors in the final model. Procrastination was not found to be a mediator but was confirmed as a significant moderator of the relationship between social media addiction and burnout. These findings underscore the urgent need for targeted interventions that focus on digital wellness and adaptive time-management strategies to support the future generation of healthcare professionals.

Ultimately, safeguarding the well-being of the next generation of healthcare professionals requires a dual-pronged approach: actively fostering digital wellness and healthier relationships with technology, while simultaneously equipping students with the psychological resilience and adaptive strategies needed to thrive in a demanding academic environment.

To support medical students’ well-being and academic performance, we recommend the following: (1) integrating digital wellness programs into university curricula; (2) offering targeted interventions to manage social media use, particularly among students with low procrastination tendencies; (3) providing training in time-management and self-regulation strategies early in medical education; and (4) establishing proactive mental health monitoring and support systems. These findings underscore the value of a dual-focus approach that addresses both digital behaviors and cognitive patterns to prevent burnout and enhance resilience among future healthcare professionals.

## Figures and Tables

**Figure 1 healthcare-13-01702-f001:**
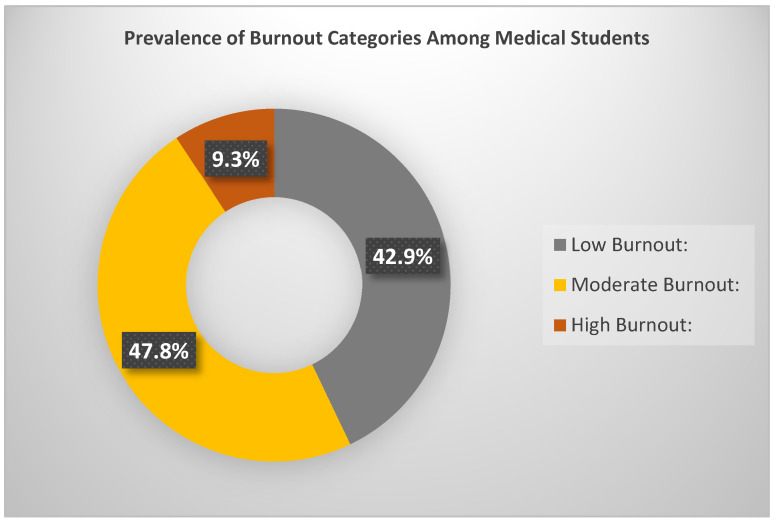
Prevalence of burnout among medical students.

**Figure 2 healthcare-13-01702-f002:**
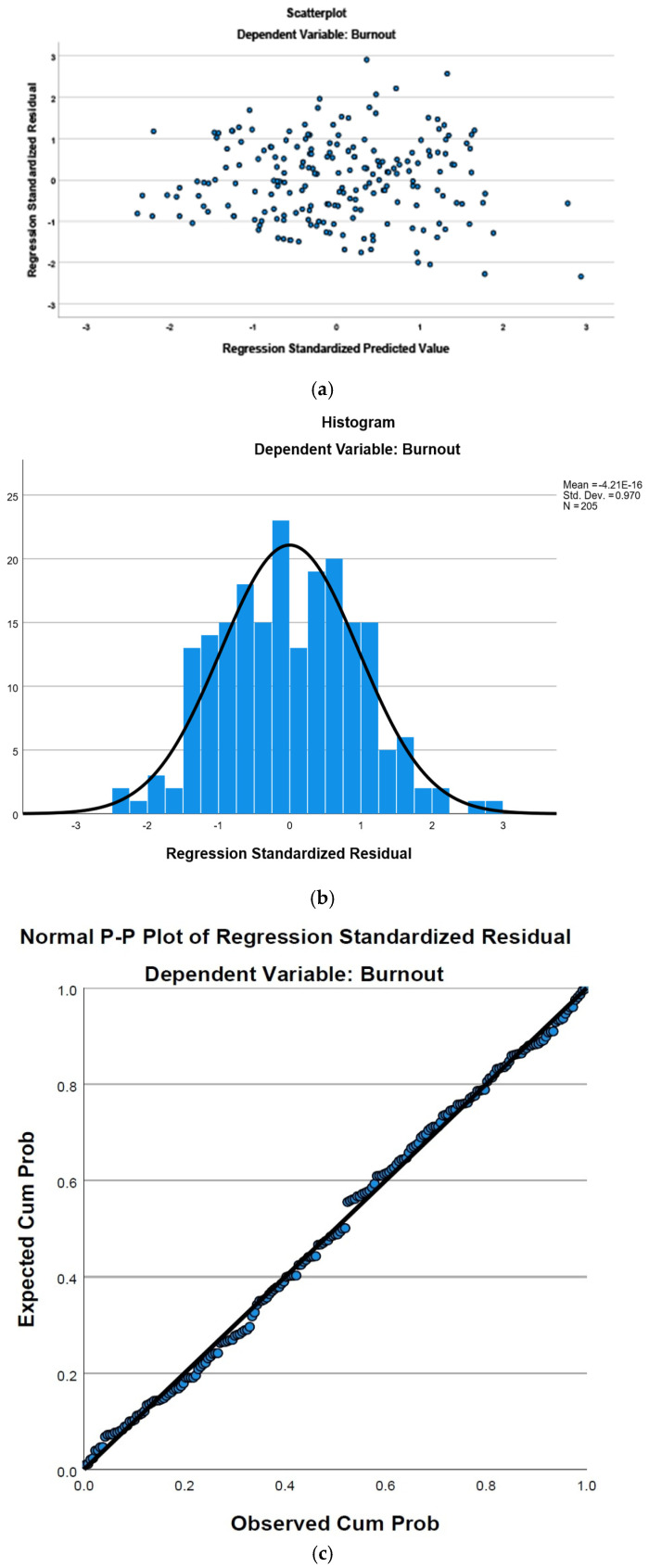
(**a**) Scatter plot of standardized residuals vs. standardized predicted values; (**b**) histogram of standardized residuals; (**c**) normal P-P plot of standardized residuals.

**Figure 3 healthcare-13-01702-f003:**
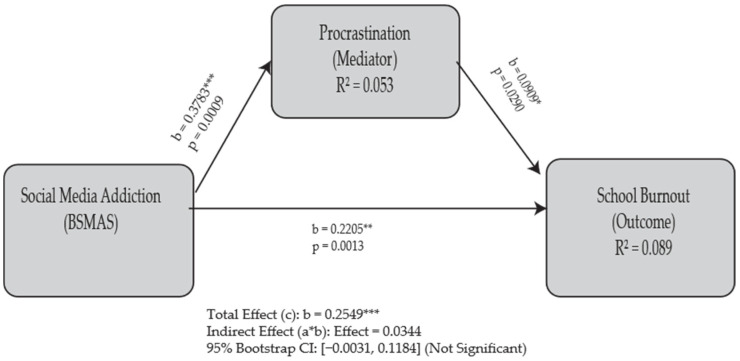
Path-analytic model of the direct and indirect effects of social media addiction (BSMAS) on school burnout through procrastination. Path coefficients represent unstandardized regression coefficients (b). The indirect effect was tested using bootstrapping and was found to be non-significant. * *p* < 0.05, ** *p* < 0.01, *** *p* < 0.001.

**Figure 4 healthcare-13-01702-f004:**
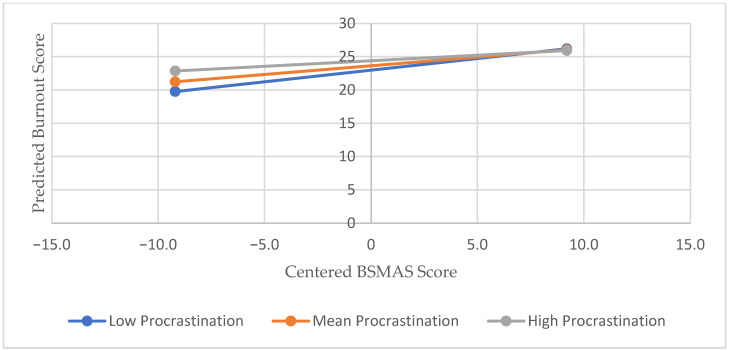
Moderation of the BSMAS-burnout relationship by procrastination.

**Table 1 healthcare-13-01702-t001:** Continuous variables (Mean, SD), *N* = 364.

	Mean	Standard Deviation
Age (18–47 years old)	22.06	6.26
Last Annual Grade	9.22	0.77
Admission Grade	9.55	0.76
Procrastination Scale (Total Score)	47.77	12.89
Total BSMAS Score	12.49	9.25
Total Burnout Score	23.44	9.02
Exhaustion	11.11	4.02
Cynicism	6.88	3.5
Inadequacy	5.45	2.77

**Table 2 healthcare-13-01702-t002:** Categorical and ordinal variables.

	Percentage
Year of study	
1st	38.5%
2nd	31%
3rd	30.5%
Sex (F)	72.3%
Do you have close people with whom you frequently discuss your problems?	
Yes, one	38.7%
Yes, several	53.5%
I don’t have	7.8%
Do you have a partner, a person you love?	
Yes	59.9%
I had, currently I don’t	11.1%
I don’t have, I want to have	11.4%
I don’t have, I’m not interested right now	17.5%
Have you obtained special results (mentions, awards) in county or national contests or competitions?	
Yes	59.6%
No	40.4%
Do you have any hobbies that you dedicate a significant portion of your time to?	
Yes	68.5%
No	31.5%
How much time do you spend on average per day in social media applications?	
Not at all	1.5%
Within 10 min	1.5%
Between 10 min and half an hour	8%
Between half an hour and an hour	12.2%
Between an hour and two hours	23.7%
Between two and three hours	19.6%
More than 3 h	33.5%

**Table 3 healthcare-13-01702-t003:** Correlation analyses between burnout and various variables.

Variable	Type	Correlation Coefficient (r/*p*)	Sig. (2-Tailed)
Age	Pearson	−0.197	0.005
Admission Grade	Pearson	0.011	0.88
Last Annual Grade	Pearson	0.013	0.849
Procrastination Score	Pearson	0.204	0.003
BSMAS Score	Pearson	0.260	<0.001
Time Spent on Applications	Spearman	0.122	0.083
Gender	Point-Biserial (Pearson)	0.023	0.748
Hobbies	Point-Biserial (Pearson)	0.044	0.533
Achievements	Point-Biserial (Pearson)	-0.036	0.609

**Table 4 healthcare-13-01702-t004:** Independent sample *t*-test results for burnout differences by dichotomous variables.

Variable	Group	Mean	SD	t-Statistic	Sig. (2-Tailed)
Gender	Male	23.08	7.97	−0.322	0.748
	Female	23.56	9.35		
Achievements	Yes	23.69	9.32	0.513	0.609
	No	23.01	8.5		
Hobbies	Yes	23.19	9.02	−0.624	0.533
	No	24.05	9.05		

**Table 5 healthcare-13-01702-t005:** One-way ANOVA results for burnout differences by categorical variables.

Variable	Group	Mean	SD	F-Statistic	Sig.
Relationship Status	Yes	23.16	9.252	0.545	0.652
	I had, currently I don’t	22.59	9.931		
	I don’t have any, I want to have	23.24	7.867		
	I don’t have any, I’m not interested right now	25.24	8.292		
Close Friends	Yes, one	23.39	8.853	1.759	0.175
	Yes, several	23.06	9.083		
	I don’t have	28.89	8.908		

**Table 6 healthcare-13-01702-t006:** Standardized regression coefficients (beta), significance levels for the final regression model, 95% confidence intervals, and effect sizes.

Predictor	Standardized Coefficient (β)	Sig. (2-Tailed)	95% CI (Lower)	95% CI (Upper)	VIF	Effect Size Magnitude
Age	−0.123	0.129	−0.396	0.05	1.416	small
Gender	0.047	0.525	−2.067	4.038	1.168	negligible
Univ year	−0.048	0.508	−3.456	1.716	1.142	negligible
Admission grade	0.006	0.935	−1.777	1.931	1.202	negligible
Last annual grade	−0.067	0.421	−3.657	1.534	1.483	negligible
Presence of hobbies	0.034	0.630	−2.072	3.417	1.085	negligible
Relationship status	0.037	0.603	−0.807	1.385	1.104	negligible
Presence of close friends	0.045	0.536	−1.567	3.001	1.130	negligible
Academic achievements	−0.032	0.675	−3.380	2.193	1.234	negligible
Procrastination score	0.137	0.061	−0.004	0.168	1.148	small
BSMAS score	0.201	0.011	0.045	0.349	1.337	small to medium
Daily social media time	0.004	0.964	−0.953	0.997	1.383	negligible

Multicollinearity: Variance Inflation Factor (VIF) values for all predictors ranged from 1.013 to 1.337, suggesting that multicollinearity was not a major concern.

## Data Availability

The original contributions presented in this study are included in the article material. Further inquiries can be directed to the corresponding author(s).

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
