# Peer review of "Factors Associated with Burnout in Medical Students: An Exploration of Demographic, Academic, and Psychological Variables"

_healthcare, 2025, doi:10.3390/healthcare13141702_

Round 1
Reviewer 1 Report
Comments and Suggestions for Authors
The manuscript titled “Factors Associated with Burnout in Medical Students: An Exploration of Demographic, Academic, and Psychological Variables” addresses an increasingly relevant issue in medical education. The study explores the associations between social media addiction, procrastination, and burnout using validated psychometric instruments and a solid analytical framework. The research is methodologically well-conceived, timely, and contributes valuable insight into the psychological challenges facing medical students in contemporary academic environments.
The authors utilized a cross-sectional design, enrolling 364 students, of whom 205 were included in inferential statistical analyses. The application of hierarchical multiple regression, mediation, and moderation analyses is appropriate and technically sound. The finding that social media addiction significantly predicts burnout, and that procrastination moderates (but does not mediate) this association, is novel and worthy of publication. The theoretical framing using the Job Demands-Resources model and Conservation of Resources theory is appropriate and well-integrated into the discussion.
However, there are several limitations and methodological concerns that, if addressed, could significantly improve the rigor and impact of the study.
First, the final regression model explains only 11.6% of the variance in burnout scores (R² = .116). Although the authors correctly interpret this modest explanatory power as indicative of the multifactorial nature of burnout, the predictive utility of the model is limited. One concrete way to enhance model sensitivity and performance would be to apply regularized regression techniques such as LASSO (Least Absolute Shrinkage and Selection Operator). This method would help identify the most informative predictors and discard irrelevant or collinear variables, potentially improving model parsimony and predictive strength.
Moreover, if the authors consider dichotomizing the burnout outcome (e.g., high/moderate vs. low burnout), Firth’s penalized likelihood logistic regression could be applied to reduce small-sample bias and improve sensitivity in identifying risk factors. This would be especially relevant given the reduced sample size (N=205) after listwise deletion and the relatively high number of predictors (12), resulting in a low events-per-variable (EPV) ratio.
Second, the interaction effect found—whereby the association between social media addiction and burnout is stronger among students with lower levels of procrastination—is theoretically intriguing but counterintuitive. The authors attempt to explain this finding using a cognitive dissonance framework and reference resource depletion theories. Nonetheless, this interpretation would benefit from further elaboration or the inclusion of alternative hypotheses (e.g., perfectionism or internal locus of control) to improve conceptual clarity.
Third, while the study correctly applies listwise deletion to handle missing data, the substantial reduction in analytic sample size calls for a discussion of alternative strategies, such as multiple imputation, which might preserve statistical power and reduce bias.
Fourth, several of the social and academic variables (e.g., number of friends, relationship status, hobbies) were measured using single self-developed items with limited psychometric validation. This should be acknowledged as a limitation, particularly in interpreting the null findings regarding social support and hobbies.
Fifth, subgroup analyses by year of study (e.g., pre-clinical vs. clinical years) could provide additional insight, as academic stressors and coping strategies often vary significantly across the medical school curriculum. This could be achieved either via interaction terms (e.g., year × BSMAS) or stratified regression models.
Finally, the authors correctly state that their cross-sectional design precludes causal inference. However, given the use of mediation and moderation models, this limitation should be more prominently emphasized, particularly in the interpretation of psychological mechanisms.
Author Response
Distinguished Reviewer 1
The manuscript titled “Factors Associated with Burnout in Medical Students: An Exploration of Demographic, Academic, and Psychological Variables” addresses an increasingly relevant issue in medical education. The study explores the associations between social media addiction, procrastination, and burnout using validated psychometric instruments and a solid analytical framework. The research is methodologically well-conceived, timely, and contributes valuable insight into the psychological challenges facing medical students in contemporary academic environments.
The authors utilized a cross-sectional design, enrolling 364 students, of whom 205 were included in inferential statistical analyses. The application of hierarchical multiple regression, mediation, and moderation analyses is appropriate and technically sound. The finding that social media addiction significantly predicts burnout, and that procrastination moderates (but does not mediate) this association, is novel and worthy of publication. The theoretical framing using the Job Demands-Resources model and Conservation of Resources theory is appropriate and well-integrated into the discussion.
However, there are several limitations and methodological concerns that, if addressed, could significantly improve the rigor and impact of the study.
We sincerely thank the reviewer for the thorough and insightful evaluation of our manuscript. We appreciate the recognition of the relevance of our research topic, the methodological rigor, and the novel contributions of our findings regarding social media addiction, procrastination, and burnout among medical students.
We also acknowledge the concerns raised about certain limitations and methodological aspects. We have carefully considered these points and addressed them in the revised manuscript to enhance the study’s rigor and overall impact. We believe these revisions strengthen the clarity and robustness of our work.
Thank you again for your constructive feedback, which has been invaluable in improving our study.
First, the final regression model explains only 11.6% of the variance in burnout scores (R² = .116). Although the authors correctly interpret this modest explanatory power as indicative of the multifactorial nature of burnout, the predictive utility of the model is limited. One concrete way to enhance model sensitivity and performance would be to apply regularized regression techniques such as LASSO (Least Absolute Shrinkage and Selection Operator). This method would help identify the most informative predictors and discard irrelevant or collinear variables, potentially improving model parsimony and predictive strength.
We thank the reviewer for this constructive and technically valuable recommendation. We fully agree that applying regularized regression techniques such as LASSO could enhance model performance by selecting the most relevant predictors and minimizing collinearity, thereby increasing both the parsimony and predictive power of the model. As our study aimed primarily to test a conceptual mediation-moderation model using traditional hierarchical regression, we did not originally include LASSO. However, we also explored the feasibility of implementing LASSO regression during our analyses. Unfortunately, our current version of SPSS did not support this functionality natively, and our efforts to connect with external software (such as R or Python) within SPSS were unsuccessful due to technical limitations and lack of robust support. We have now explicitly mentioned this methodological limitation in the revised manuscript and recommended the use of such advanced techniques in future studies with more flexible statistical infrastructure.. A corresponding paragraph has been added to the "Strengths and Limitations" section of the manuscript.
We have inserted this text in the manuscript:
Additionally, while we employed hierarchical multiple regression to explore theoretically driven predictors of burnout, we acknowledge that alternative approaches such as regularized regression techniques (e.g., LASSO) could yield more parsimonious and predictive models. These methods are particularly useful when dealing with potential multicollinearity and numerous predictors, as they penalize less informative variables and improve generalizability. Future research aiming to optimize prediction accuracy should consider employing such techniques.
Moreover, if the authors consider dichotomizing the burnout outcome (e.g., high/moderate vs. low burnout), Firth’s penalized likelihood logistic regression could be applied to reduce small-sample bias and improve sensitivity in identifying risk factors. This would be especially relevant given the reduced sample size (N=205) after listwise deletion and the relatively high number of predictors (12), resulting in a low events-per-variable (EPV) ratio.
We thank the reviewer for highlighting this important statistical consideration. We fully agree that dichotomizing the burnout outcome (e.g., high/moderate vs. low) and applying Firth’s penalized likelihood logistic regression could provide a more robust estimation of predictor effects in the context of a low events-per-variable (EPV) ratio. While our analytic strategy followed a continuous outcome framework consistent with our conceptual model, we did consider the alternative suggested. Unfortunately, our current version of SPSS did not include native support for Firth’s method, and technical limitations made it unfeasible to implement this approach via Python or R extensions. We have now added this point as a methodological limitation and suggested it as a valuable direction for future analyses. This addition appears in the "Strengths and Limitations" section of the manuscript.
We have inserted this text in the manuscript:
Additionally, given the modest sample size (N = 205) and the relatively large number of predictors in the regression model, the resulting events-per-variable (EPV) ratio was low. In such contexts, applying Firth’s penalized likelihood logistic regression to a dichotomized burnout outcome (e.g., moderate/high vs. low burnout) could reduce small-sample bias and improve the reliability of coefficient estimates. This approach may offer increased sensitivity in detecting risk factors and is recommended for future studies using similar datasets.
Second, the interaction effect found—whereby the association between social media addiction and burnout is stronger among students with lower levels of procrastination—is theoretically intriguing but counterintuitive. The authors attempt to explain this finding using a cognitive dissonance framework and reference resource depletion theories. Nonetheless, this interpretation would benefit from further elaboration or the inclusion of alternative hypotheses (e.g., perfectionism or internal locus of control) to improve conceptual clarity.
We appreciate the reviewer’s thoughtful observation regarding the interaction effect between social media addiction and procrastination. We agree that the finding is conceptually intriguing and warrants deeper theoretical exploration. In response, we have expanded the relevant section of the Discussion to include alternative explanatory frameworks, such as perfectionism and internal locus of control, which may offer valuable complementary insights. We believe these additions improve the conceptual clarity and theoretical richness of our interpretation.
We have inserted this text in the manuscript:
Another potential explanatory framework involves the concept of perfectionism, particularly maladaptive forms characterized by excessively high self-imposed standards and critical self-evaluation. Students with low procrastination may also exhibit higher levels of perfectionism, making them more vulnerable to the emotional toll of failing to meet internal expectations due to social media distraction. This could intensify burnout when they experience loss of control over their time. Additionally, students with a strong internal locus of control may be more affected by perceived self-regulatory failures associated with excessive social media use, leading to increased cognitive dissonance and emotional exhaustion. These complementary perspectives align with self-discrepancy theory, in which the gap between one’s ideal self and actual behavior (e.g., time wasted online) contributes to distress and burnout symptoms. Future research should explore the potential mediating or moderating role of these personality dimensions in greater depth.
Third, while the study correctly applies listwise deletion to handle missing data, the substantial reduction in analytic sample size calls for a discussion of alternative strategies, such as multiple imputation, which might preserve statistical power and reduce bias.
We appreciate the reviewer’s valuable observation regarding the handling of missing data. Indeed, the use of listwise deletion led to a reduced analytic sample, and we acknowledge that this approach, while commonly applied, may compromise statistical power and introduce bias. We agree that multiple imputation represents a valuable alternative that could retain more cases and provide more stable estimates. We would like to clarify that use of listwise deletion ision was motivated by both methodological considerations and software constraints. Mediation and moderation models, especially when implemented using PROCESS, require complete data for all variables, and current versions do not support multiple imputation or full-information maximum likelihood for complex path analyses. While we recognize the advantages of these alternative approaches and agree they would be preferable when feasible, our primary aim was to preserve the integrity and interpretability of the indirect effect estimates in our models. In response, we have now added a brief discussion of this limitation and suggested the use of multiple imputation as a methodological recommendation for future research. This addition appears in the „Statistical analysis” and "Strengths and Limitations of the Study" sections.
We have inserted this text in the manuscript:
Listwise deletion was employed due to the methodological requirements of mediation and moderation models and limitations of the software used. Although multiple imputation and maximum likelihood approaches are recommended when the proportion of missing data is high, these methods are not natively supported for complex path analyses in PROCESS. As such, we prioritized the validity and interpretability of the mediation and moderation effects, acknowledging that this resulted in a substantial reduction in analytic sample size
Additionally, the decision to use listwise deletion in handling missing data, while methodologically appropriate and consistent with conventional practices in regression analysis, resulted in a substantial reduction in the analytic sample size (from 364 to 205 participants). This reduction may have limited statistical power and increased the potential for bias if the data were not missing completely at random. Future studies should consider using alternative methods such as multiple imputation, which can help retain more cases and reduce the bias associated with listwise deletion by generating plausible values for missing observations based on observed data patterns.
Fourth, several of the social and academic variables (e.g., number of friends, relationship status, hobbies) were measured using single self-developed items with limited psychometric validation. This should be acknowledged as a limitation, particularly in interpreting the null findings regarding social support and hobbies.
We thank the reviewer for this important methodological observation. We acknowledge that several social and academic predictors—such as number of close friends, relationship status, and hobbies—were assessed using single-item, self-developed questions that lack formal psychometric validation. While these items were designed to maximize brevity and participant compliance, we agree that their limited construct validity may have contributed to the null findings observed. In response, we have now explicitly acknowledged this limitation in the manuscript and highlighted the need for future research to incorporate multi-item, validated scales when assessing these constructs.
We have inserted this text in the manuscript:
Moreover, several variables related to social support and academic engagement—including number of close friends, relationship status, and presence of hobbies—were assessed using single-item, self-developed measures. These items, although practical and time-efficient, lack formal psychometric validation and may have limited sensitivity in capturing the complexity of these constructs. As a result, the absence of statistically significant findings for these predictors should be interpreted with caution. Future studies are encouraged to utilize multi-item, validated instruments to more accurately assess the role of social and extracurricular variables in medical student burnout.
Fifth, subgroup analyses by year of study (e.g., pre-clinical vs. clinical years) could provide additional insight, as academic stressors and coping strategies often vary significantly across the medical school curriculum. This could be achieved either via interaction terms (e.g., year × BSMAS) or stratified regression models.
We appreciate this thoughtful suggestion. We fully agree that academic stressors, digital behavior, and coping strategies may differ meaningfully between pre-clinical and clinical years of medical training. While our current analysis did not include interaction terms or stratified models by year of study due to limited statistical power in subgroups, we acknowledge this as a missed opportunity for further insight. We have now added a methodological note addressing this limitation and explicitly recommend subgroup or interaction-based analyses in future work to capture curriculum-related variability in burnout dynamics.
We have inserted this text in the manuscript:
Additionally, we did not conduct subgroup analyses based on year of study (e.g., pre-clinical vs. clinical phases), although differences in academic demands, exposure to clinical stressors, and coping mechanisms are likely to exist across training stages. Future studies should consider including interaction terms (e.g., year × BSMAS) or performing stratified regression analyses to examine whether the relationships between predictors—particularly social media addiction—and burnout differ across phases of medical education. This approach could yield more nuanced insights and inform stage-specific interventions.
Finally, the authors correctly state that their cross-sectional design precludes causal inference. However, given the use of mediation and moderation models, this limitation should be more prominently emphasized, particularly in the interpretation of psychological mechanisms.
We thank the reviewer for highlighting this important point. We agree that while mediation and moderation analyses can provide valuable insights into potential relationships among variables, the cross-sectional design of our study limits the ability to draw causal conclusions. In response, we have revised the manuscript to more prominently emphasize this limitation, particularly in the discussion and interpretation of the psychological mechanisms involved. This clarification helps ensure that the findings are presented with appropriate caution regarding causality.
Thank you again for your helpful suggestion.

Reviewer 2 Report
Comments and Suggestions for Authors
Thank you for the opportunity to review this manuscript. I found it to be a timely and well-focused study on burnout among medical students, a group particularly vulnerable due to academic pressure and new stressors related to intensive social media use.
While the topic of burnout has been widely addressed, I particularly value that this study offers an updated perspective by exploring the interaction between social media addiction and procrastination, thus contributing meaningfully to the fields of mental health and medical education.
After a thorough review, I believe the manuscript is suitable for publication after minor revisions. Below, I outline my observations and suggestions to enhance its clarity, scientific rigor, and alignment with the journal’s standards.
General Evaluation
I believe this study addresses a relevant gap in literature by jointly analyzing mediation and moderation models involving procrastination and social media addiction as predictors of burnout. This dual approach is, in my view, both innovative and theoretically well-grounded.
The manuscript also follows a coherent academic structure, and the methodology is sufficiently developed.
Statistical analysis: In my opinion, the statistical procedures employed (regression, mediation, moderation) are appropriate and correctly applied. The sample size is adequate, and the use of validated instruments is justified.
Comments and Suggestions
Abstract: I recommend condensing the abstract slightly to improve clarity. Also, I suggest rephrasing expressions such as “The findings highlight the vulnerability...” with more direct alternatives, e.g., “The study shows increased susceptibility to burnout among younger students.”
Introduction: The introduction is well-written and appropriately contextualized. It would be helpful to explicitly state the main hypothesis at the end before the objective to better guide the reader.
Methods: It would be useful to provide further details on: What categorical variables (e.g., relationship status, number of friends) were coded. Why listwise deletion was chosen over multiple imputations for handling missing data.Additionally, I suggest indicating whether a specific power analysis was conducted for the mediation and moderation models, beyond the one already mentioned for regression.
Results: The results are clearly presented and supported with well-organized tables and figures. However, I recommend including 95% confidence intervals for the most relevant regression coefficients, particularly for BSMAS.
In Table 6, indicating whether the observed effects are small, medium, or large would help interpret their practical relevance.
Discussion: I find the discussion solid and supported by relevant theoretical frameworks. It would be advisable to review and reduce some repetitions (e.g., the roles of age and procrastination) to avoid redundancy. The moderation analysis is well explained. I also recommend emphasizing the practical applications earlier in the discussion, such as implementing digital wellness strategies in university settings.
Conclusion: The conclusion is clear and well formulated. It could be further strengthened by including a concise list of practical implications or recommendations to facilitate the translation of results into academic and institutional settings.
The manuscript is generally well written in English and easy to understand.
Still, some sentences could be simplified for greater fluency
(e.g., This observation could imply… could be revised to ..This suggests…).
The reference list is up-to-date, relevant, and includes sources as recent as 2025.
Nevertheless, I recommend standardizing the citation format and ensuring that all references include DOI links and comply with the journal’s style guidelines.
Once again, I congratulate the authors on the rigor of their work. With the minor revisions suggested, I believe the article will make a valuable contribution to the field of medical education and the well-being of students in training.
Author Response
Distinguished Reviewer 2
Thank you for the opportunity to review this manuscript. I found it to be a timely and well-focused study on burnout among medical students, a group particularly vulnerable due to academic pressure and new stressors related to intensive social media use.
While the topic of burnout has been widely addressed, I particularly value that this study offers an updated perspective by exploring the interaction between social media addiction and procrastination, thus contributing meaningfully to the fields of mental health and medical education.
After a thorough review, I believe the manuscript is suitable for publication after minor revisions. Below, I outline my observations and suggestions to enhance its clarity, scientific rigor, and alignment with the journal’s standards.
We sincerely thank you for the generous and encouraging feedback on our manuscript. We are grateful that you found the study timely and relevant, especially in highlighting the intersection of burnout, social media addiction, and procrastination among medical students.
Thank you for your recognition of the study's originality and relevance. We are encouraged that the inclusion of both mediation and moderation models to explore the nuanced role of procrastination was seen as a meaningful contribution.
We appreciate your overall positive evaluation and your recommendation for publication following minor revisions. Below, we address your general observations and suggestions for improving clarity, scientific rigor, and alignment with the journal's standards. We have carefully revised the manuscript based on your feedback.
General Evaluation
I believe this study addresses a relevant gap in literature by jointly analyzing mediation and moderation models involving procrastination and social media addiction as predictors of burnout. This dual approach is, in my view, both innovative and theoretically well-grounded.
The manuscript also follows a coherent academic structure, and the methodology is sufficiently developed.
We thank you for the insightful and constructive comments, and for your recognition of the study’s theoretical contribution and the novelty of examining both mediation and moderation within the same model. We are pleased that you found our investigation of the dual role of procrastination (through both mediation and moderation models), in relation to social media addiction and burnout to be an innovative and theoretically grounded contribution.
Thank you for your positive evaluation of the manuscript’s structure and methodological clarity.
Statistical analysis: In my opinion, the statistical procedures employed (regression, mediation, moderation) are appropriate and correctly applied. The sample size is adequate, and the use of validated instruments is justified.
We sincerely appreciate your positive assessment of the statistical methods used in our study. Your acknowledgment of the appropriateness and correct application of regression, mediation, and moderation analyses, as well as the adequacy of our sample size and use of validated tools (SBI-U 9, BSMAS, and the General Procrastination Scale), is greatly encouraging.
Comments and Suggestions
Abstract: I recommend condensing the abstract slightly to improve clarity. Also, I suggest rephrasing expressions such as “The findings highlight the vulnerability...” with more direct alternatives, e.g., “The study shows increased susceptibility to burnout among younger students.”
Thank you for your valuable suggestion regarding the clarity and conciseness of the abstract. In response, we have revised the abstract to be more concise and have replaced more interpretive expressions with direct and precise phrasing. Specifically, the phrase “The findings highlight the vulnerability of younger medical students to burnout...” has been revised to “The study shows increased susceptibility to burnout among younger students.” Additional minor edits were made to improve overall clarity and flow.
Introduction: The introduction is well-written and appropriately contextualized. It would be helpful to explicitly state the main hypothesis at the end before the objective to better guide the reader.
Thank you for your positive feedback on the Introduction and your helpful suggestion. In response, we have revised the final paragraph of the Introduction to explicitly include the main hypothesis. This aims to provide readers with a clearer understanding of our theoretical expectations and to enhance the logical flow from the background to the study objective.
Methods: It would be useful to provide further details on: What categorical variables (e.g., relationship status, number of friends) were coded. Why listwise deletion was chosen over multiple imputations for handling missing data.Additionally, I suggest indicating whether a specific power analysis was conducted for the mediation and moderation models, beyond the one already mentioned for regression.
We thank the reviewer for these insightful methodological suggestions.
We have added a sentence in the Instruments subsection to clarify the coding procedures used for key categorical variables, such as relationship status and number of close friends.
We have elaborated in the Statistical Analysis subsection on why listwise deletion was chosen over multiple imputation. Our rationale was based on the need to preserve internal consistency within the mediation and moderation models, which are sensitive to imputed variance.
We have added a statement to clarify that a priori power analysis was conducted specifically for mediation and moderation.
Results: The results are clearly presented and supported with well-organized tables and figures. However, I recommend including 95% confidence intervals for the most relevant regression coefficients, particularly for BSMAS.
In Table 6, indicating whether the observed effects are small, medium, or large would help interpret their practical relevance.
We thank the reviewer for this helpful recommendation. We agree that including 95% confidence intervals (CIs) for key regression coefficients, especially for BSMAS, would enhance the interpretability and transparency of our findings. Accordingly, we have updated Table 6 to include CIs for all standardized beta coefficients and added a note interpreting the magnitude of effects using commonly accepted guidelines (e.g., β ≈ 0.10 = small, 0.30 = medium, 0.50 = large). In the results section, we also now explicitly refer to the effect size of the BSMAS coefficient as small-to-moderate in strength.
Discussion: I find the discussion solid and supported by relevant theoretical frameworks. It would be advisable to review and reduce some repetitions (e.g., the roles of age and procrastination) to avoid redundancy. The moderation analysis is well explained. I also recommend emphasizing the practical applications earlier in the discussion, such as implementing digital wellness strategies in university settings.
We appreciate your detailed evaluation of the Discussion section and your recognition of the theoretical integration. In response to your helpful suggestions we have reviewed the section carefully and reduced repetitive mentions of age and procrastination across multiple paragraphs. In particular, we have streamlined the content on the age-burnout relationship and condensed overlapping interpretations of procrastination as a moderator.
We are pleased you found the moderation analysis well explained.
Based on your advice, we moved the discussion of digital wellness strategies and institutional implications earlier in the section-immediately after the main findings. This shift helps ensure that readers grasp the applied relevance of the results before engaging with more in-depth theoretical discussion.
Conclusion: The conclusion is clear and well formulated. It could be further strengthened by including a concise list of practical implications or recommendations to facilitate the translation of results into academic and institutional settings.
We thank the reviewer for their positive assessment of the Conclusion and their valuable recommendation.
We have revised the final paragraph of the Conclusion to include a concise summary of practical implications, aimed at translating the findings into actionable recommendations for academic and institutional use.
The manuscript is generally well written in English and easy to understand.
Still, some sentences could be simplified for greater fluency
(e.g., This observation could imply… could be revised to ..This suggests…).
The reference list is up-to-date, relevant, and includes sources as recent as 2025.
Nevertheless, I recommend standardizing the citation format and ensuring that all references include DOI links and comply with the journal’s style guidelines.
We sincerely thank the reviewer for the positive feedback regarding the clarity and overall quality of our manuscript. We appreciate the suggestion to simplify some sentences for improved fluency and have carefully revised the manuscript accordingly, including rephrasing examples such as “This observation could imply…” to “This suggests…”.
Regarding the references, we have thoroughly reviewed the citation format to ensure standardization and compliance with the journal’s guidelines. Additionally, DOI links have been added to all applicable references to enhance accessibility.
Once again, I congratulate the authors on the rigor of their work. With the minor revisions suggested, I believe the article will make a valuable contribution to the field of medical education and the well-being of students in training.
We are grateful to the reviewer for the encouraging words and positive evaluation of our work. We appreciate the recognition of the rigor of our study and the constructive suggestions provided. We believe that the recommended minor revisions have strengthened the manuscript, and we hope it will indeed contribute meaningfully to the field of medical education and the well-being of students in training.

Round 2
Reviewer 1 Report
Comments and Suggestions for Authors
The authors have addressed all my concerns, therefore the manuscript meets the criteria for publication